# Coping with Cybervictimization: The Role of Direct Confrontation and Resilience on Adolescent Wellbeing

**DOI:** 10.3390/ijerph16244893

**Published:** 2019-12-04

**Authors:** Antonella Brighi, Consuelo Mameli, Damiano Menin, Annalisa Guarini, Francesca Carpani, Phillip T. Slee

**Affiliations:** 1Faculty of Education, Free University of Bolzano, 39100 Bolzano, Italy; 2College of Education Psychology and Social Work, Flinders University, Adelaide, SA 5001, Australia; 3Department of Educational Sciences, University of Bologna, 40126 Bologna, Italy; 4Department of Psychology, University of Bologna, 40126 Bologna, Italy

**Keywords:** cybervictims, cyberbullying, resilience, coping strategies, emotional symptoms

## Abstract

Background. Recent studies have consistently identified the negative consequences of cyberbullying on adolescent mental health. Nevertheless, not all cybervictims are alike, and in the last few years some evidence has appeared indicating that faced with cyberbullying, victims may manifest different emotional outcomes. In this study, we explored whether cybervictim resilience fully or partially mediates the effects of cybervictimization and whether a confrontational coping strategy impacts emotional symptoms. Methods. The study was carried out with a sample of 474 high school students equally distributed between males and females. Data were collected using a questionnaire comprising four measures assessing cybervictimization, direct confrontation coping strategy, resilience and emotional symptoms. Results. Structural equation modelling indicated that the effects of cybervictimization and confrontational coping strategy on emotional symptoms were mediated by resilience, with cybervictimization showing a positive effect while direct confrontation a negative effect. Cybervictimization also showed a positive direct effect on emotional symptoms. Conclusions. These results are presented in light of their implications for designing effective interventions able to protect and promote adolescents’ psychological wellbeing.

## 1. Introduction

The internet and associated online activities are now a ubiquitous aspect of young people’s daily life. Along with positive aspects, there are risks, including being victimized online. Estimates of the prevalence of online victimization vary greatly affected by definitional and measurement issues amongst other factors. In order to reduce methodological bias, using the same procedure and the same tools, a recent study [1] described the epidemiology of cyberbullying across Europe (Bulgaria, Cyprus, France, Greece, Hungary, Italy, Poland and Spain), revealing that the prevalence of cybervictimization was around 16% with some differences among countries (e.g., higher rate in Bulgaria and Hungary and less prevalence in Spain). The prevalence of cybervictimization described among Italian adolescents was about 18% for males and 20.5% for females.

Research has consistently identified the consequences of cyberbullying: in relation to the victims it is associated with a range of negative emotional and behavioural outcomes for adolescents including suicidal ideation and, in extreme cases, attempts and completion by victims [2,3]. Similar results were reported in a cross-sectional study of data from a very large sample of participants recruited for the International Health Behavior in School-Aged Children survey [4]: Vieno and collaborators showed that adolescents who reported experiencing cybervictimization were nearly twice as likely to experience psychological and somatic symptoms than were their non-victimized peers; this effect increased from occasional to frequent cybervictimization.

This evidence was confirmed by a meta-analysis by Gini, Card and Pozzoli [5], where both traditional and cyber forms of victimization showed unique relations with internalizing problems.

Nevertheless, responses to cyberbullying by victims vary widely but only a few studies have considered those who experience cyberbullying and who apparently do not suffer strong negative effects. Recent studies on the emotional consequences of bullying and cyberbullying have pointed out that even faced with the same kind of aggression, victims may manifest different emotional outcomes [6,7,8]. Different constructs have been considered in explaining such individual differences, e.g., the type of cyberbullying suffered (public or private), the intensity, the anonymity of the perpetrators and the comorbidity with other forms of bullying experienced. Moreover, perceived self-esteem, social support [9], coping style [10] and resilience [11,12,13] have been identified as protective factors against cyberbullying, entailing different strategies by the victim to counter it compared with face to face bullying.

The attention to protective factors, both individual and environmental, have been emphasized by several authors [14,15,16]. Protective factors are elements that may compensate, buffer or moderate the negative effects of exposure to risk. Resilience theory, in particular, seems a fruitful approach since it calls attention to processes that contribute to a better than expected functioning or “bounce back” from adversity [17,18].

To date, resilience has been rather overlooked in the studies on cybervictimization despite its potential role for promoting positive outcomes even in the presence of harmful online experiences. In cyberspace, resilience has been conceptualized as “being able to deal with a negative experience online, i.e., not remaining passive but displaying problem-solving coping strategies in order to protect oneself from future harm” [19] (p. 60). The few existent studies [11,12,20] on the role of resilience in moderating the impact of cybervictimization are promising. Gianesini and Brighi [11], for example, explored the mediational role of resilience amongst cyberbullying and cybervictimization, and mental wellbeing, reporting that high levels of resilience were able to predict healthy adjustment and fewer psychosomatic complaints for both victims and perpetrators.

Considering the dynamic and situational nature of resilience, Rutter [15,21] suggested that this dimension should be investigated only as complementary to two other factors. These include risk factors—since resilience, by nature, is activated only in conjunction with life-threatening events—and protective factors, where “protection may derive from what people do to deal with stress or adversity” [21] (p. 8). According to Rutter’s suggestion, coping strategies can be thus conceptualized as pathways to resilience [13] (p. 123), as they represent the “efforts to adapt to stress or other disturbances by a stressor or adversity in order to protect oneself from the psychological harm of risky experiences” [19] (p. 61).

The literature has identified a number of models purporting to explain how young people can defend themselves against bullying and cyberbullying [10,13]. Among the different models available in the literature, according to Raskauskas and Huynh [13] one of the most widely used refers to the transactional model of stress and coping (TMSC) [22]. The TMSC distinguishes between two main coping categories, i.e., Emotion- vs. Problem-focused. Emotion-focused coping is centred on emotions and is designed to manage the negative feelings associated with stressful events [23,24]. Problem-focused coping, conversely, is centred on the problem itself and it is aimed at modifying and solving a challenging situation.

Among the various problem-focused coping strategies—which range from technical solutions such as blocking contacts to retaliation [25,26,27]—direct confrontation may represent a particularly relevant strategy because it emphasizes the victim’s attempt to assume a resilient and active position with respect to the bully. Specifically, this strategy involves seeking a direct but not aggressive confrontation with the bully in order to stop his/her behaviour [28], and it is widely used in cases where the source of the bullying is known [23,29], with up to 60% of adolescents using it [25]. Although some studies have had mixed results regarding its effectiveness [30], a study by Machackova and collaborators [31] showed that in the less serious cases of online harassment, 71% of the victims believe that direct confrontation is emotionally helpful and 66% of them declare that it is effective in stopping bullying. The same authors concluded that “children might be encouraged to use confrontation when the situation is not severe” [31] (p. 9). In addition, problem-focused coping strategies that deal with the stressor directly have been associated with fewer health complaints compared with coping techniques aimed at evading the stressor [24].

As previously noted, prior research has highlighted the potentially crucial role of resilience in buffering the effects of harmful experience. However, only a few empirical investigations systematically focused on the role of this construct in the context of cybervictimization. In addition, to our knowledge, no study has adopted Rutter’s suggestions [15,21] investigating resilience alongside both risk (i.e., being cyberbullied) and protective (i.e., coping strategies) factors such as direct confrontation, as it is considered an adaptive response at least in cases of mild cyberbullying. Therefore, the research outlined in this paper is intended to address the shortcomings identified in the reviewed literature focusing on the question of whether victims’ resilience fully or partially mediates the relationship between cybervictimization and disposition towards direct confrontation, on the one hand, and emotional symptoms, on the other. This model is expected to account for some of the variability in the affective response among cybervictims. Moreover, since previous research has found evidence of gender differences concerning cybervictimization [32], coping strategies [33], resilience [34] and emotional issues [35,36], in this study we controlled for the effect of this dimension on all the study variables. Since the study sample was characterized by a wide range for age (13–19 years), we also controlled for the potential effects of age.

## 2. Methods

### 2.1. Participants

Participants were 474 Italian secondary school students (241 males, 231 females and two who did not indicate gender), coming mostly from middle class families (19.2% of participants’ parents completed compulsory education, 43.35% completed high school and 29.14% had a university qualification). Participants were enrolled in three academic and technical secondary schools located in Northern Italy. The average age of the participants was 15.28 years (SD = 1.23, range 13–19). They were almost equally divided into students attending the 9th (n = 241, 50.8%) and the 11th (n = 232, 48.9%) grades. Most of the participants had Italian as their mother tongue (n = 401; 84.6%), while the remaining students spoke fluent Italian. A multiple regression highlighted no significant effect (α = 0.05) of having completed the survey on any of the latent variables included in the study, indicating that the missing-at-random assumption was plausible.

### 2.2. Questionnaire

Cybervictimization was assessed using an 8-item scale from the European Cyberbullying Intervention Project Questionnaire (ECIPQ) [9,37,38] concerning experiences of victimization online. The experience of victimization covers different behaviours (e.g., “Someone spread rumours about me online”) including direct and indirect aggression and social exclusion. Students were asked to answer each item on a 5-point Likert scale (0 = Never; 1 = One or two times; 2 = Monthly; 3 = Weekly; 4 = Several times in the last week). We defined occasional cybervictimization as that which occurred one or two times in the last six months, according to the literature [8,35]. In our sample, Cronbach’s alpha for the scale was 0.83.

The disposition towards direct confrontation was assessed using a 4-item subscale (e.g., “Tell the bully to stop what they are doing”, “Stick up for myself”) from an adapted and translated version of the Coping with Bullying Questionnaire [39]. Participants were asked to indicate their responses to real or hypothetical situations on each item on a 5-point Likert scale (from 1 = Never to 5 = Always). In our sample, Cronbach’s alpha for the scale was 0.75.

Psychological resilience was assessed using the Resilience Scale (RS-14) [40]. In line with other authors [41,42] we opted for a one-dimension solution of the latent construct “Resilience”. This 14-item scale, evaluated on a 5-point Likert scale (from 1 = Absolutely disagree” to 5 = Absolutely agree; e.g., “I feel proud that I have accomplished things in my life”, “I am determined”) showed a good reliability in our sample, with a Cronbach’s alpha of 0.81.

We used the Strengths and Difficulties Questionnaire (SDQ) [43,44] to measure emotional distress. In particular, for this paper we took into account only the Emotional Symptoms subscale, which comprises five items on physical symptoms and worries (e.g., “often unhappy, depressed or tearful”). Students were asked to answer to each item on a 3-point Likert scale (1 = Not true, 2 = In part true, 3 = True). For this subscale, in our sample the Cronbach’s alpha obtained was 0.76.

### 2.3. Procedures

The online questionnaire was filled in during school hours. The research was introduced to the students by a trained researcher as a survey on cybervictimization and a brief explanation was provided. It took approximately 15 to 20 min to complete the questionnaire. During the survey, teachers remained in the classrooms in order to clarify any questions or problems.

### 2.4. Ethics

The study protocol met the ethical guidelines for protection of human participants, including adherence to the legal requirements of the country and was formally approved by the Bioethics Committee, University of Bologna. Both participants’ parents provided their informed written consent for the student’s participation in the study, data analysis and for anonymous data publication. For students over 18, an informed consent form was delivered to them and signed before they were enrolled in the study.

### 2.5. Statistical Analysis

We used structural equation modeling to test the hypothesis that resilience mediates the effects of cybervictimization and of disposition towards direct confrontation on emotional symptoms, controlling for gender differences.

Goodness of fit was assessed using the comparative fit index (CFI), the Tucker–Lewis index (TLI), the root mean square error of approximation (RMSEA) and the standardized root mean square residual (SRMR). Index values of 0.90 or higher indicated an acceptable fit for CFI and TLI, and values lower than respectively 0.06 and 0.08 reflected an adequate fit for RMSEA and SRMR [45,46].

Since our survey data were coded as ordered, we considered the survey items as ordinal variables [47,48]. As a consequence, instead of the classic ML estimator, we adopted the weighted least squares with means and variance adjusted (WLSMV), a robust version of the diagonally weighted least squares (DWLS) estimator specifically designed for ordinal data [49,50,51], with Satorra–Bentler correction for skewed data. Confirmatory factor analysis (CFA) and structural equation modeling (SEM) were carried out using Lavaan version 0.5-23.1097 in R version 3.4.1. The same measurement model specified for CFA was retained in SEM analysis.

## 3. Results

### 3.1. Descriptive Statistics and Correlation Matrix

As shown in Table 1, in reporting the prevalence of each type of cybervictimization, all different behaviours were present and some of them had an incidence rate higher than 20% (e.g., “Someone said nasty things to me or threatened me using texts or online messages”, “Someone said nasty things about me to others either online or through text messages”, “Someone spread rumours about me on the internet”). In addition, 18.4% of participants reported only one type of cybervictimization, 15.2% reported two types, 9.7% three types and 15.2% four or more different types of victimization, with a total of 58% of respondents involved in cybervictimization during the last six months.

Table 2 shows Pearson correlations and descriptive data for the study variables. Descriptive data are presented in relation to the total sample for male and female groups. Cybervictimization showed a negative correlation with resilience *r*(398) = −0.21, *p* < 0.001 and a positive correlation with emotional symptoms, *r*(398) = 0.29, *p* < 0.001. Disposition to direct confrontation showed a positive correlation with resilience, *r*(398) = 0.26, *p* < 0.001. Resilience showed a negative correlation with emotional symptoms, *r*(398) = −0.33, *p* < 0.001.

### 3.2. Mediation Analysis Through Structural Equation Modeling

A CFA was performed in order to test the measurement model, highlighting an adequate fit: *CFI* = 0.921, *TLI* = 0.914, *RMSEA* = 0.046 (90% CI 0.041; 0.051), *SRMR* = 0.073. Factor loadings were all significantly different from zero. Next, the hypothesized structural model (see Figure 1) was fitted. The tested model included resilience as a mediator of the effects of cybervictimization and disposition to direct confrontation on emotional symptoms. In addition, we tested the effects of gender and age on each of the latent variables (cybervictimization, direct confrontation, resilience and emotional symptoms).

Figure 2 shows the tested model, with standardized regression coefficients. For the sake of simplicity gender, age and other measured variables (i.e., single items included in the measurement model) were not reported in the figure. The model fit was acceptable: *CFI* = 0.953, *TLI* = 0.955, *RMSEA* = 0.051 (90% CI 0.047; 0.055), *SRMR* = 0.073. The effect of cybervictimization on emotional symptoms was partly mediated by resilience (indirect effect, *β* = 0.038, 95% CI 0.011 to 0.065, *p* = 0.006), since cybervictimization negatively affected resilience (*β* = −0.156, 95% CI −0.268 to −0.044, *p* = 0.006), and resilience negatively affected emotional symptoms (*β* = −0.243, 95% CI −0.355 to −0.131, *p* < 0.001). In addition, cybervictimization showed a direct effect on emotional symptoms (*β* = 0.300, 95% CI 0.184 to 0.417, *p* < 0.001). By contrast, the effect of disposition to direct confrontation on emotional symptoms was fully accounted for by the mediation of resilience (*β* = −0.090, 95% CI −0.140 to −0.039, *p* < 0.001), since the direct effect was not significant (*β* = −0.067, 95% CI −0.179 to 0.045, *p* = 0.240). In particular, disposition to direct confrontation increased resilience (*β* = 0.369, 95% CI 0.270 to 0.469, *p* < 0.001).

Moreover, gender differences were highlighted, with males reporting less cybervictimization (*β* = −0.197, 95% CI −0.312 to −0.082, *p* < 0.001), more resilience (*β* = 0.231, 95% CI 0.126 to 0.336, *p* < 0.001) and less emotional symptoms (*β* = −0.434, 95% CI −0.532 to −0.335, *p* < 0.001), while the disposition to direct confrontation was not influenced by gender (*β* = −0.087, 95% CI −0.194 to 0.020, *p* = 0.116). Age highlighted a positive effect on disposition to direct confrontation (*β* = 0.136, 95% CI 0.033 to 0.238, *p* = 0.011), while it did not affect emotional symptoms (*β* = 0.013, 95% CI −0.082 to 0.107, *p* = 0.792), resilience (*β* = −0.004, 95% CI −0.106 to 0.098, *p* = 0.936) nor cybervictimization (*β* = −0.047, 95% CI −0.160 to 0.66, *p* = 0.416).

## 4. Discussion

The present study confirms the pervasiveness of cybervictimization in the context of Italian secondary schools, in line with previous studies [1,37,52]. Moreover, our findings support the view that cybervictimization—even if occasional—can affect psychological and emotional wellbeing [7,8].

Nevertheless, as discussed in the introduction to our study, research identified a range of different emotional reactions to cybervictimization [7,8,11]. Beginning with these considerations, our research question was to investigate whether victims’ resilience fully or partially mediates the relationship between cybervictimization and disposition towards direct confrontation on the one hand, and emotional symptoms, on the other.

Our results show that the victims of frequent cyberattacks suffered both directly in terms of emotional symptoms and indirectly in terms of a reduction in their general resilience. This means that cybervictimization can be a threatening experience for adolescents’ mental wellbeing and given its pervasiveness should be taken seriously whenever reported. Importantly, our research seems to indicate that a proactive coping strategy such as direct confrontation exerts a protective effect on emotional symptoms, an effect which is fully mediated by resilience. Therefore, coping seems to be a pathway to resilience, as suggested by Raskauskas and Huynh [13], at least in relation to direct confrontation, as its enactment improves resilience. As Bandura [53] pointed out, the most powerful work in resilience is promoting personal agency and confidence in one’s ability to develop resilience. As such, direct confrontation might represent an agentic choice where actions and learned habits have the potential to grow and expand over time [53,54]. This means that, as for other “resilience practices” [55], adolescents can learn “what to do” in stressful situations, overcoming the boundaries of personal dispositions as well as those imposed by environmental determinants. In this regard, further research should explore the role of other coping strategies in relation to resilience, as well as to emotional issues and to other outcomes (e.g., externalizing behaviours, social problems, etc.).

Our results are consistent with Rutter’s hypothesis—i.e., that resilience can be interpreted as an outcome of the interplay of both risk and protective factors—and point out a processual view of resilience [13,15,21], which in turn affects the entity of emotional symptoms. This is an interesting and innovative finding—with theoretical and practical implications for school-based interventions—since the different impact of both cybervictimization and direct confrontation on resilience and on emotional symptoms has not been tested before. Hereafter, it would be important to test the relations between resilience and potential protective and risk factors, taking into account more complex models which consider both individual and contextual variables.

In addition, our results add some interesting considerations on the effect of gender and age. On the one hand, our data confirm that females are more at risk for cybervictimization, suggesting a relatively greater involvement in girls as cybervictims [56,57]. Our results also confirm a lack of resilience and more emotional symptoms among girls, as frequently reported in existing literature [32,36]. On the other hand, the present study adds a possible explanation of the relation among these variables, suggesting that a high risk for cybervictimization in females is combined with a lack of resilience, resulting in more severe emotional symptoms. With respect to the role of age, the age-related increase in direct confrontation might be influenced by an improvement of technical skills in the use of internet or by a greater familiarity in dealing with online harassment, as suggested by other studies [58] and stressing the idea that coping strategies towards bullying and cyberbullying tend to increase with age.

## 5. Conclusions

Due to the exploratory nature of this study, further research is needed in order to overcome some methodological limitations that may affect the interpretation of our results. First, our findings are limited with respect to the sample size and the socio-cultural context related to the school settings where the study was conducted, so evidence from other contexts and countries is necessary. Moreover, a bigger sample could allow for confirmation of our results, especially regarding those relatively small effects that may vary across different populations and conditions. Second, our study relies on self-report data which can influence the subjects’ responses according to social desirability issues, although it should be noted that in the case of cybervictimization individual self-reports are likely to provide the most accurate data. Third, this is a cross-sectional study investigating processes which may occur over time; therefore, longitudinal data could shed light on the dynamic process involving stressful events and psychological adjustment. Finally, additional research is needed in order to analyse possible gender differences in the dynamics of resilience in the face of cybervictimization—e.g., fitting separate models for males and females.

Notwithstanding these limitations, our study has contributed some new insight into the role of resilience in response to aggression taking place in virtual contexts. As suggested by Estrada et al. [59], we need to integrate the study of resilience across domains (such as face to face bullying and other forms of aggression) to fully understand its nature and the processes that can foster it in the face of different stressors. This would contribute new insight into the development of effective interventions for cybervictimization.

## Figures and Tables

**Figure 1 ijerph-16-04893-f001:**
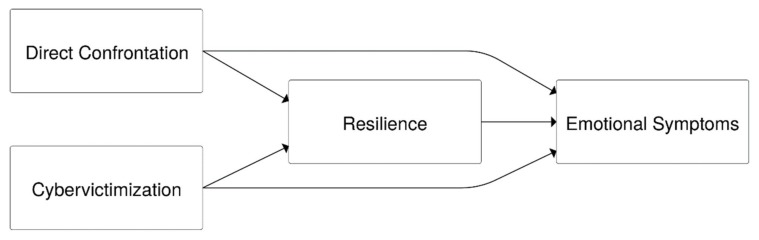
Proposed path model.

**Figure 2 ijerph-16-04893-f002:**
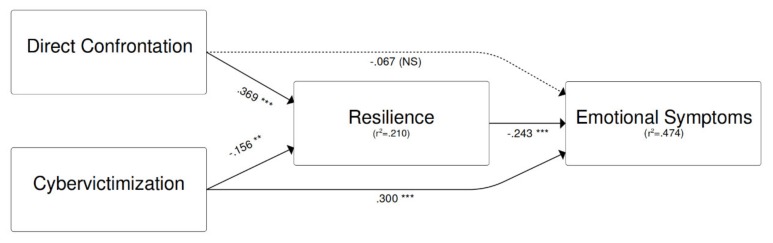
Final path model.

**Table 1 ijerph-16-04893-t001:** Prevalence of different types of cyberbullying in the sample.

Item	Prevalence in Percentage				
	1 or 2 Times	Monthly	Weekly	More Often	Total
1. Someone said nasty things to me or threatened me using texts or online messages	23.1	2.3	1.1	2.5	29.0
2. Someone said nasty things about me to others either online or through text messages	27.5	2.5	1.3	1.3	32.6
3. Someone hacked into my account and stole personal information (e.g., through email or social networking accounts)	14.2	1.7	0.2	0.4	16.5
4. Someone pretended to be me, hacking into my account or creating a fake one	9.7	0.6	0.4	0.6	11.4
5. Someone posted personal information about me online	12.3	1.7	0.4	0.8	15.3
6. Someone posted embarrassing videos or pictures of me online or altered pictures or videos I had posted online	14.0	1.5	1.3	0.8	17.6
7. I was excluded or ignored by others in a social networking site or internet chat room	12.7	2.3	0.6	1.3	16.9
8. Someone spread rumours about me on the internet	16.5	2.8	0.6	1.5	21.4

**Table 2 ijerph-16-04893-t002:** Correlation matrix and descriptive statistics for the study variables.

Study Variables	1	2	3	4
1. Direct confrontation	‒			
2. CB victimization	−0.036	‒		
3. Resilience	0.259 ***	−0.208 ***	‒	
4. Emotional symptoms	−0.063	0.287 ***	−0.327 ***	‒
Total sample [M (SD)]	3.33 (1.09)	0.27 (0.43)	3.72 (0.53)	1.67 (0.52)
Males [M (SD)]	3.22 (1.17)	0.23 (0.49)	3.82 (0.48)	1.43 (0.37)
Females [M (SD)]	3.44 (0.99)	0.32 (0.37)	3.61 (0.56)	1.92 (0.53)

Note. Descriptive statistics are calculated on mean values for each scale; *** *p* < 0.001.

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
