# Peer review of "Coping with Cybervictimization: The Role of Direct Confrontation and Resilience on Adolescent Wellbeing"

_ijerph, 2019, doi:10.3390/ijerph16244893_

Round 1

Reviewer 1 Report

The study presented in this manuscript focuses on a topic of great importance these days: cyberbullying. In fact, the performance of this study is well justified by the authors in the introduction, as this was aimed at filling existing gaps in research on this topic. However, there some minor concerns that could be addressed before publication:

The provision of some rates about the prevalence of cyberbullying would give greater emphasis on the importance of the topic under study. It seems quite strange that emotional symptoms were only assessed through a five-item questionnaire. The authors state in the Procedure section that “The online questionnaire was filled in during school hours”. However, participants’ age ranged from 13 to 19 years. In this connection, potential age effects on results should also have been explored, as the authors assessed both adolescents and young adults. Some differences in results depending on participants’ gender are mentioned in the Results section, but these are not discussed later. The authors state that “the research outlined in this paper is intended to address the shortcomings identified in the reviewed literature focusing on the question of whether victims’ resilience fully or partially mediates the relationship between cybervictimization and disposition towards direct confrontation, on the one hand, and emotional symptoms, on the other.” However, the Discussion largely focuses on the impact of those variables on resilience, instead of the opposite.

Author Response

We wish to thank Reviewer #1 for his/her appreciation to the topic of our study and for his/her meaningful and important suggestions. Based on his/her comments, it was possible for us to improve the paper and strengthen it in several parts. Below we list the revisions we made based on his/her suggestions.

Q1: The provision of some rates about the prevalence of cyberbullying would give greater emphasis on the importance of the topic under study.

RE: We have added some information about the prevalence of cybervictimization across Europe, as described by a very recent study carried out in Bulgaria, Cyprus, France, Greece, Hungary, Italy, Poland, and Spain (lines 34-42, see below).

“The internet and associated on-line activities are now a ubiquitous aspect of young people’s daily life. Along with positive aspects there are risks including being victimised on-line. Estimates of the prevalence of online victimization vary greatly affected by definitional and measurement issues amongst other factors. In order to reduce methodological bias, using the same procedure and the same tools, a recent study [1] has described the epidemiology of cyberbullying across Europe (Bulgaria, Cyprus, France, Greece, Hungary, Italy, Poland, and Spain), revealing that the prevalence of cybervictimization was around the 16% with some differences among Countries (e.g., higher rate in Bulgaria and Hungary and less prevalence in Spain). The prevalence of cybervictimization described among Italian adolescents was about 18% for males and 20.5% for females.”

Q2: It seems quite strange that emotional symptoms were only assessed through a five-item questionnaire.

RE: We certainly understand the concern expressed by reviewer #1, and we would like to reply him/her by giving some more information about this scale. As we explained in the method section, the SDQ complete scale was administered to our participants, but in this study we considered only the Emotional symptoms sub-scale, as it is more focused on Internalizing and psychosomatic complaints. The SDQ questionnaire is very widely used, translated into more than eighty languages (https://sdqinfo.org/py/sdqinfo/b0.py) with normative values in 10 Countries, including Italy. In addition, the SDQ questionnaire was used in more the 4499 publications from 100 Countries, as indicated in the Questionnaire website (https://sdqinfo.org/a0.html).

In addition, the items from this subscale are particularly fit to describe the kinds of symptoms extensively reported in literature as a consequence of cybervictimization (cfr. the meta-analysis by Gini, Card and Pozzoli, 2018).

Q3: The authors state in the Procedure section that “The online questionnaire was filled in during school hours”. However, participants’ age ranged from 13 to 19 years. In this connection, potential age effects on results should also have been explored, as the authors assessed both adolescents and young adults.

RE: We thank the reviewer for his/her suggestion. Based on his/her comment, we have entered AGE into the SEM model as an exogenous variable. This variable was found to positively affect the disposition to direct confrontation. Modified coefficients have been reported according to the new model (see lines 220-242 below).

“Figure 2 shows the tested model, with standardized regression coefficients. For the sake of simplicity gender, age and other measured variables (i.e., single items included in the measurement model) were not reported in the figure. The model fit was acceptable: CFI = .953, TLI = .955, RMSEA = .051 (90% CI .047; .055), SRMR = .073. The effect of cybervictimization on emotional symptoms was partly mediated by resilience (indirect effect, β = 0.038, 95% CI 0.011 to 0.065, p = .006), since cybervictimization negatively affected resilience (β = -0.156,, 95% CI -0.268 to -0.044, p = .006), and resilience negatively affected emotional symptoms (β = -0.243, 95% CI -0.355 to -0.131, p < .001). In addition, cybervictimization showed a direct effect on emotional symptoms (β = 0.300, 95% CI 0.184 to 0.417, p < .001). By contrast, the effect of disposition to direct confrontation on emotional symptoms was fully accounted for by the mediation of resilience (β = -0.090, 95% CI -0.140 to -0.039, p < .001), since the direct effect was not significant (β = -0.067, 95% CI -0.179 to 0.045, p = .240). In particular, disposition to direct confrontation increased resilience (β = 0.369, 95% CI 0.270 to 0.469, p < .001). Moreover, gender differences were highlighted, with males reporting less cybervictimization (β = -0.197, 95% CI -0.312 to -0.082, p < .001), more resilience (β = 0.231, 95% CI 0.126 to 0.336, p < .001), and less emotional symptoms (β = -0.434, 95% CI -0.532 to -0.335, p < .001), while the disposition to direct confrontation was not influenced by gender (β = -0.087, 95% CI -0.194 to 0.020, p = .116). Age highlighted a positive effect on disposition to direct confrontation (β = 0.136, 95% CI 0.033 to 0.238, p = .011), while it did not affect emotional symptoms (β = 0.013, 95% CI -0.082 to 0.107, p = .792), resilience (β = -0.004, 95% CI -0.106 to 0.098, p = .936) nor cybervictimization (β = -0.047, 95% CI -0.160 to 0.66, p = .416).”

Age-related differences were also discussed in the discussion section (lines 282-286, see below).

“With respect to the role of age, the age-related increase in direct confrontation might be influenced by an improvement of technical skills in the use of internet or by a greater familiarity in dealing with online harassment, as suggested by other studies [58] stressing the idea that coping strategies towards bullying and cyberbullying tend to increase with age. ”

Q4: Some differences in results depending on participants’ gender are mentioned in the Results section, but these are not discussed later.

RE: As suggested by the Reviewer, gender differences were discussed in the discussion section (lines 276-282, see below).

“In addition, our results add some interesting considerations on the effect of gender and age. On the one hand, our data confirm that females are more at risk for cybervictimization, suggesting a relatively greater involvement in girls as cybervictims [56,57]. Our results also confirm a lack of resilience and more emotional symptoms among girls, as frequently reported in existing literature [32,36]. On the other hand, the present study adds a possible explanation on the relation among these variables, suggesting that a high risk for cybervictimisation in females is combined with a lack of resilience, resulting in more severe emotional symptoms.”

Q5: The authors state that “the research outlined in this paper is intended to address the shortcomings identified in the reviewed literature focusing on the question of whether victims’ resilience fully or partially mediates the relationship between cybervictimization and disposition towards direct confrontation, on the one hand, and emotional symptoms, on the other.” However, the Discussion largely focuses on the impact of those variables on resilience, instead of the opposite. 

RE: We thank the reviewer for pointing out the shortcomings of the discussion. As suggested, we have extensively revised this section in order to make as clear as possible the mediation model that was tested, i.e. with disposition to direct confrontation and cybervictimization as exogenous variables, resilience as the mediator, and emotional symptoms as the endogenous variable. Therefore, in our model, resilience was affected by disposition to direct confrontation and cybervictimization and affected emotional symptoms (lines 245-286).

Reviewer 2 Report

The paper examines the relationship between cybervicitimization and well-being on the basis of an occasional sample. An important addition to existing literature is that the mediating role of resilience is investigated. Before the manuscript can be published, at least the following points should be adressed:
1. The title of the article ("Not all the cybervictims are aleike") is misleading. It suggests that different groups of young people will be examined in the article (e.g. on the basis of latent class analyses). However, this is not the case; rather, classical correlational analyses are carried out. The title should therefore be reworded.
2. In the article, the terms "cyberbullying" and "cybervictimization" are used almost synonymously. However, the authors only examine cybervictimization. If cyberbullying were to be investigated, a different operationalization of the dependent variable would have to be carried out (at least weekly victimization).
3. The authors formulate on p. 6: "The effect of cybervictimization on emotional symptoms was partly mediated by resilience". However, it can be doubted that a correlation of .038 (-.156 * -.243) in the sample of 474 respondents is actually significant. The mediation effect should therefore be tested for significance by bootstrapping, especially since it is the central hypothesis of the manuscript.
4. In a first step, the authors calculate confirmatory factor analyses but then they include only manifest variables, not latent variables in the structural equation model. Why is no complete structural equation model (including measurement models) specified?

Author Response

We wish to thank Reviewer #2 for his/her appreciation to the topic of our study and for his/her meaningful and important suggestions. Based on his/her comments, it was possible for us to improve the paper and strengthen it in several parts. Below we list the revisions we made based on his/her suggestions.

Q1: The title of the article ("Not all the cybervictims are alike") is misleading. It suggests that different groups of young people will be examined in the article (e.g. on the basis of latent class analyses). However, this is not the case; rather, classical correlational analyses are carried out. The title should therefore be reworded.

RE: As suggested by reviewer #2, in order to make the title clearer we changed it to “Coping with Cybervictimization: The Role of Direct Confrontation and Resilience on Adolescent Wellbeing”. In this way it was possible for us to highlight in the title the important role of direct confrontation and resilience on wellbeing.

Q2: In the article, the terms "cyberbullying" and "cybervictimization" are used almost synonymously. However, the authors only examine cybervictimization. If cyberbullying were to be investigated, a different operationalization of the dependent variable would have to be carried out (at least weekly victimization).

RE: As suggested, the manuscript was checked and corrected in order to make sure that it was always clear that we were referring to cybervictimization.

Q3: The authors formulate on p. 6: "The effect of cybervictimization on emotional symptoms was partly mediated by resilience". However, it can be doubted that a correlation of .038 (-.156 * -.243) in the sample of 474 respondents is actually significant. The mediation effect should therefore be tested for significance by bootstrapping, especially since it is the central hypothesis of the manuscript.

RE: We thank the reviewer for this important observation. In answering, we should clarify that the mentioned coefficient does not refer to a correlation, but rather to the standardized regression path within the specified SEM model, after controlling for all other effects.

Bootstrapping has been proposed as an alternative approach to multivariate non-normality when distribution-free estimation methods are not available (see Nevitt & Hancock, 2001*). However, as stated in the manuscript (lines 177-178), since our survey data were coded as ordered, we considered the survey items as ordinal variables. As a consequence, instead of the classic ML estimator, we adopted the weighted least squares with means and variance adjusted (WLSMV), a robust version of the diagonally weighted least squares (DWLS) estimator, specifically designed for ordinal data, which does not assume normality for measured variables (Beauducel & Herzberg, 2006; Li, 2016; Rhemtulla, Brosseau-Liard, & Savalei, 2012). Moreover, in the revised version of the manuscript we adopted Satorra-Bentler correction for skewed data (lines 180-183). We preferred this approach to the option suggested by the reviewer, considering our sample size and multivariate non-normal distribution, because bootstrapping was found to produce non-reliable results with N<500 accompanied by severely non-normal distributions (Nevitt & Hancock, 2001). As for the small effect size, this might be due to different moderators or variations relative to various sub-populations that should be investigated by adopting a larger sample size (Gini, Card, & Pozzoli, 2018). Therefore, we introduced a new sentence in the conclusion stating this point (lines 292-293). In addition, 95 % Confidence Intervals were added in the Results section, in order to provide a more accurate description of results. In general, we still think that even a small effect (often observed in mediation models) on emotional symptoms, which consist of serious emotional difficulties, may deserve attention.

*Nevitt & Hancock, 2001, Performance of bootstrapping approaches to model test statistics and parameter standard error estimation in structural equation modeling. Structural equation modeling, 8(3), 353-377.

Lines 177 – 183: “Since our survey data were coded as ordered, we considered the survey items as ordinal variables [47,48]. As a consequence, instead of the classic ML estimator, we adopted the weighted least squares with means and variance adjusted (WLSMV), a robust version of the diagonally weighted least squares (DWLS) estimator, specifically designed for ordinal data [49–51], with Satorra-Bentler correction for skewed data. Confirmatory factor analysis (CFA) and structural equation modeling (SEM) were carried out using Lavaan version 0.5-23.1097 in R version 3.4.1. The same measurement model specified for CFA was retained in SEM analysis.”

Lines 292-293: “Moreover, a bigger sample could allow to confirm our results, especially regarding those relatively small effects that may vary across different populations and conditions.”

Q4. In a first step, the authors calculate confirmatory factor analyses but then they include only manifest variables, not latent variables in the structural equation model. Why is no complete structural equation model (including measurement models) specified?

RE: We thank the reviewer for pointing out that this was not clear in the original manuscript. We specified in lines 182-183 and 214-215 of the revised manuscript that SEM analysis included the same measurement model as CFA.

“ The same measurement model specified for CFA was retained in SEM analysis.”

“In addition, we tested the effects of gender and age on each of the latent variables (cybervictimization, direct confrontation, resilience, and emotional symptoms).”